# Designing Counterfactual Generators using Deep Model Inversion

**Jayaraman J. Thiagarajan**
Lawrence Livermore National Laboratory
jjayaram@llnl.gov

**Vivek Narayanaswamy**
Arizona State University
vnaray29@asu.edu

**Deepta Rajan**
IBM Research AI
r.deepta@gmail.com

**Jason Liang**
Stanford University
jialiang@stanford.edu

**Akshay Chaudhari**
Stanford University
akshaysc@stanford.edu

**Andreas Spanias**
Arizona State University
spanias@asu.edu

## Abstract

Explanation techniques that synthesize small, interpretable changes to a given image while producing desired changes in the model prediction have become popular for introspecting black-box models. Commonly referred to as counterfactuals, the synthesized explanations are required to contain discernible changes (for easy interpretability) while also being realistic (consistency to the data manifold). In this paper, we focus on the case where we have access only to the trained deep classifier and not the actual training data. While the problem of inverting deep models to synthesize images from the training distribution has been explored, our goal is to develop a deep inversion approach to generate counterfactual explanations for a given query image. Despite their effectiveness in conditional image synthesis, we show that existing deep inversion methods are insufficient for producing meaningful counterfactuals. We propose DISC (Deep Inversion for Synthesizing Counterfactuals) that improves upon deep inversion by utilizing (a) stronger image priors, (b) incorporating a novel *manifold consistency* objective and (c) adopting a progressive optimization strategy. We find that, in addition to producing visually meaningful explanations, the counterfactuals from DISC are effective at learning classifier decision boundaries and are robust to unknown test-time corruptions.

## 1 Introduction

With the growing need for deploying deep black-box models into critical decision-making, there is an increased emphasis on explainability methods that can reveal intricate relationships between data signatures (e.g., image features) and predictions. In this context, the so-called counterfactual (CF) explanations [1] that synthesize small, interpretable changes to a given image while producing desired changes in model predictions to support user-specified hypotheses (e.g., progressive change in predictions) have become popular. Though counterfactual explanations provide more flexibility over conventional techniques, such as feature importance estimation [2, 3, 4, 5, 6], by exploring the vicinity of a query image, an important requirement to produce meaningful counterfactuals is to produce discernible local perturbations (for easy interpretability) while being realistic (close to the underlying data manifold). Consequently, existing approaches rely extensively on pre-trained generative models to synthesize plausible counterfactuals [1, 7, 8, 9, 10]. By design, this ultimately restricts their utility to scenarios where one cannot access the training data or pre-trained generative models, for example, due to privacy requirements commonly encountered in many practical applications.

In this paper, we focus on the problem where we have access only to trained deep classifiers and not the actual training data or generative models. Synthesizing images from the underlying data distribution

by inverting a deep model, while not requiring access to training data, is a well investigated topic of research. For e.g., Deep Dream [11] synthesizes class-conditioned images by manipulating a noisy image directly in the space of pixels (or more formally *Image Space Optimization* (ISO)) constrained by image priors such as total variation [12] to regularize this ill-posed inversion. However, Deep Dream is known to produce images that look unrealistic, often very different from the training images, thus limiting their use in practice. Consequently, Yin *et al.*, proposed *DeepInversion* [13] that performs image synthesis in the latent space of a pre-trained classifier (*Latent Space Optimization* (LSO)) and leverages layer-specific statistics (from batchnorm [14]) to constrain the images to be consistent with the training data distribution. This was showed to produce higher-quality images, particularly in the context of performing knowledge distillation [15] using the synthesized images.

**Proposed Work.** In contrast, this work aims to develop a deep model inversion approach that generates counterfactual explanations by exploring the vicinity of a given query image, instead of synthesizing an arbitrary realization from the entire image distribution. As illustrated in the example in Figure 1, existing deep inversion methods are ineffective when natively adopted for counterfactual generation. Due to use of weak priors, and the severely ill-posed nature of the problem, it introduces irrelevant pixel manipulations that easily satisfy the desired change in prediction. Hence, we propose DISC (Deep Inversion for Synthesizing Counterfactuals) that improves upon conventional deep model inversion by utilizing: (i) stronger image priors through the use of deep image priors [16]

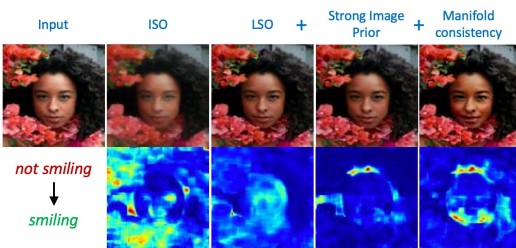

Figure 1: We propose DISC, a deep model inversion approach for query-based CF generation. Using a strong image prior (INR in this example) and our manifold consistency constraint, along with a progressive optimization strategy, DISC introduces discernible yet semantically meaningful changes (rightmost) to the query image.

(DIP) and implicit neural representations [17] (INR); (ii) a novel *manifold consistency* objective that ensures the counterfactual remains close to the underlying manifold; and (iii) a progressive optimization strategy to effectively introduce discernible, yet meaningful, changes to the query image.

From Figure 1, we find that our approach produces meaningful image manipulations, in order to change the prediction to the *smiling* class, while other deep inversion strategies cannot. Using empirical studies, we show that DISC consistently produces visually meaningful explanations, and that the counterfactuals from DISC are effective at learning model decision boundaries and are robust to unknown test-time corruptions.

**Our Contributions.**

1. A general framework to produce counterfactuals on-the-fly using deep model inversion;

2. Novel objectives to ensure consistency to the data manifold. We explore two different strategies based on direct error prediction [18, 19] and deterministic uncertainty estimation [20];

3. A progressive optimization strategy to introduce discernible changes to a given query image, while satisfying the manifold consistency requirement;

4. A *classifier discrepancy* metric to evaluate the quality of counterfactuals;

5. Empirical studies using natural image and medical image classifiers to demonstrate the effectiveness of DISC over a variety of baselines and ablations.

## 2  Related Work

**Image Synthesis from Classifiers.** Inverting a pre-trained deep model is a popular strategy for image synthesis in scenarios where there is no access to the underlying training data. While deep model inversion-based methods such as *Deep Dream* [11] and *DeepInversion* [13] have been successful in generating class-conditioned images, there have been other extensions to such approaches. For example, Dosovitskiy *et al.* [21] proposed to invert representations of a pre-trained CNN to obtain insights about what a deep classifier has learned. On similar lines, Mahendran *et al.* [12, 22] addressed the problem of pre-image recovery, which in essence attempts to recover an arbitrarily

encoded representation (in the latent space of a classifier) to a realization on the (unknown) image manifold and to enable model diagnosis. Ulyanov *et al.* [16] improved upon this ill-posed inversion by utilizing a strong image prior in the form of *deep image priors* (DIP). They also explored the related problem of activation-maximization [16], where the goal is to generate an image that maximizes the activation of a given output neuron in a pre-trained classifier, and demonstrated the effectiveness of DIP. Despite the effectiveness of the deep model inversion methods for image synthesis, we find that such methods cannot be natively adopted for CF generation and are insufficient for producing meaningful pixel manipulations to a given query image.

**Counterfactual Generation.** Existing methods extensively rely on generative models to provide counterfactuals that explain the decisions of a black-box model. Examples including CounteR-GAN [23], Counterfactual Generative Networks (CGNs) [24] and the methods reported in [7, 8], have clearly demonstrated the use of generative models in synthesizing high-quality CFs for any user-specified hypothesis on the predictions. However, this requirement of access to training data or generative models can be infeasible in practical scenarios, for e.g., restrictions arising due to privacy requirements. In contrast, our approach formulates the problem of counterfactual generation using deep model inversion, and can produce meaningful counterfactuals using only the trained classifier.

## 3 Proposed Approach

In this section, we describe our approach for counterfactual generation that improves upon deep model inversion, and introduce a new classifier discrepancy metric for evaluating CFs. There are four key components that are critical to designing classifier-based counterfactual generators: (i) choice of metric for *semantics preservation*; (ii) choice of image priors to regularize image synthesis; (iii) *manifold consistency* to ensure that the synthesized counterfactual lies close to the true data manifold; and (iv) progressive optimization strategy to introduce gradual meaningful changes to a query image. Figures 2a and 2b provide overview of DISC implemented with deep image prior and INR-based for regularizing the counterfactual optimization process.

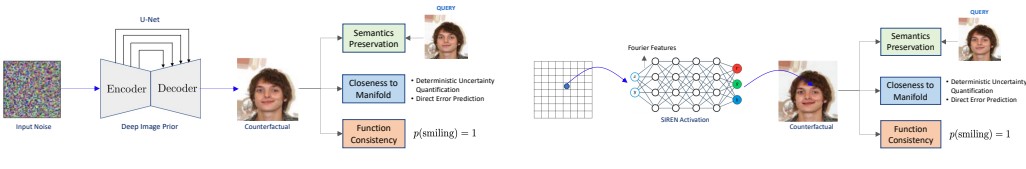

(a) Deep Image Prior          (b) Implicit Neural Representations

Figure 2: Overview of the proposed approach. For a given query, DISC trains an image generator, which can be implemented using a deep image prior (DIP) or coordinate-based neural representations (INR), based on three key objectives: (i) semantics preservation; (ii) manifold consistency; and (iii) function consistency.

In its simplest form, for a given query x, a counterfactual explanation can be obtained as follows:

$$\arg \min_{\bar{x}} d(\bar{x}, x) \quad \text{s.t.} \quad \mathcal{F}(\bar{x}) = \bar{y}, \ \bar{x} \in \mathcal{M}(x), \tag{1}$$

where $\bar{x} = \mathcal{C}(x)$ is a counterfactual explanation for x, $\mathcal{F}$ is a pre-trained classifier model, $\mathcal{M}$ denotes the data manifold and $\bar{y} = y + \delta$ is the desired change in the prediction. The metric $d(.,.)$ measures the discrepancy between the query image and the counterfactual (i.e., semantics preservation).

### 3.1 Choice of Metric for Semantics Preservation

We can measure the discrepancy between a query and its CF explanation, $d(\bar{x}, x)$, in the pixel space or in the latent space of the classifier. We now describe the high-level formulation of deep model inversion-based CF generation.

**a. Image Space Optimization (ISO).** ISO for counterfactual generation involves the ill-posed optimization of an input $\bar{x}$ directly in the pixel space to generate an image semantically similar to the query x, while being consistent with a user-hypothesis on the prediction, $\bar{y}$. Strategies such as Deep Dream [11] perform ISO to synthesize artistic variations of images. Mathematically,

$$\arg \min_{\bar{x}} d(\bar{x}, x) + \mathcal{R}(\bar{x}) \quad \text{s.t.} \quad \mathcal{F}(\bar{x}) = \bar{y}. \tag{2}$$

where $\mathcal{R}(\bar{x})$ is a suitable image prior to regularize the optimization.

**b. Latent Space Optimization (LSO).** LSO refers to the ill-posed problem of inverting an arbitrarily encoded representation from the latent space of a deep classifier to a realization on the (unknown) image manifold. Let $\Psi_l(.)$ denote the $l^{th}$ differentiable layer of the deep classifier. Then, counterfactual generation using LSO can be mathematically formulated as

$$\arg\min_{\bar{x}} \sum_l d(\Psi_l(\bar{x}), \Psi_l(x)) + \mathcal{R}(\bar{x}) \quad \text{s.t.} \quad \mathcal{F}(\bar{x}) = \bar{y}. \tag{3}$$

Approaches such as DeepInversion [13] perform LSO for conditional image synthesis (though with distribution-level comparison instead of our sample-level comparison) and achieve visually superior images when compared to ISO approaches.

## 3.2 Choice of Image Priors

As observed from (2) and (3), the choice of the regularizer or image prior $\mathcal{R}(.)$ is central towards regularizing and tractably solve this challenging inverse problem. A variety of image priors have been proposed in the literature, and we investigate the following in this work.

**a. Total Variation + $\ell_2$.** *Total variation* [22] (TV) is a popular regularizer that encourages images to contain piece-wise constant patches while the $\ell_2$ *norm* regularizes the range and energy of the image to remain within a given interval. The TV norm and the $\ell_2$ norm are given by:

$$\mathcal{R}_{TV}(\bar{x}) = \sum_{i,j} \sqrt{(\bar{x}_{i,j+1} - \bar{x}_{i,j})^2 + (\bar{x}_{i+1,j} - \bar{x}_{i,j})^2}; \quad \mathcal{R}_{\ell_2}(\bar{x}) = \sqrt{\sum_{i,j} \|\bar{x}_{i,j}\|^2} \tag{4}$$

**b. Deep Image Priors (DIP).** A *Deep Image Prior* (DIP) [16] leverages the structure of an untrained, carefully tailored convolutional neural network (e.g., U-Net [25]) to generate images and solve a variety of ill-posed restoration tasks in computer vision. DIP has been found to produce high-quality reconstructions, based on the key insight that the structure of the network itself can act as a regularizer. Consequently, the synthesized image is re-parameterized in terms of the weights $\theta$ of the prior model $f_\theta$ i.e $\bar{x} = f_\theta(\mathbf{z})$.

**c. Implicit Neural Representations (INR).** We also considered an alternative approach based on INR, which provide a cheap and convenient way to learn a continuous mapping from the image coordinates to the pixel values (RGB). While they have been found to be effective for image/volume rendering and designing generative models [26], we explore their use in deep model inversion (details in appendix). We build upon two key results to design our INR-based counterfactual generators:

(i) *Fourier mapping*: Based on NTK (neural tangent kernel) theory, [27] showed that using Fourier mapping can recover high-frequency features in low-dimensional coordinate-based image reconstruction. Hence, we use a Fourier feature mapping $z$ to featurize $2-$D input coordinates $v \in [0,1]^2$ before passing them through a coordinate-based MLP:

$$z(v) = [a_1 \cos(2\pi b_1^T v), a_1 \sin(2\pi b_1^T v), \cdots].$$

Using a set of randomly chosen sinusoids, this maps the input points to the surface of a high-dimensional hypersphere. Training the MLP network on these embedded points corresponds to kernel regression with the stationary composed NTK $h_{\text{NTK}} \circ h_z$, where $h_{\text{NTK}}$ denotes the neural tangent kernel corresponding to the MLP;

(ii) *SIREN activation*: In [17], Sitzmann *et al.* showed that periodic activation functions are better suited for recovering natural images and their derivatives, when compared to standard activation functions. More specifically, SIREN uses a sinusoid activation $\Phi(x) = \sin(\mathbf{W}x + b)$. We find that using both a Fourier mapping coupled with SIREN activation leads to a very strong image prior.

## 3.3 Manifold Consistency

A key constraint in (1) that is not included in the formulations in (2), (3) is the manifold consistency, and interestingly, this is not inherently satisfied in deep model inversion. Consequently, even with a strong image prior, it can synthesize images that do not belong to the original data distribution. This

can be particularly challenging when producing CFs that represent change in class labels, wherein one expects patterns specific to a target class to be emphasized. To address this challenge, we extend the formulation in (3) (and equivalently (2)) to include a manifold consistency constraint, that is defined directly based on the classifier, without assuming access to class-specific statistics in the latent space. More specifically,:

$$\arg \min_{\bar{x}} \lambda_1 \sum_l d(\Psi_l(\bar{x}), \Psi_l(x)) + \lambda_2 \mathcal{L}_{mc}(\bar{x}; \mathcal{F}) + \lambda_3 \mathcal{L}_{fc}(\mathcal{F}(\bar{x}), \bar{y}). \tag{5}$$

The first term for *semantics preservation* is same as that of (3) and is used to ensure that the inherent semantics of the query image is retained in the generated counterfactual (implemented as the $\ell_2$ error). The second term $\mathcal{L}_{mc}$ (*manifold consistency*) penalizes solutions that do not lie close to the data manifold and is designed by assuming access only to the classifier $\mathcal{F}$. The final loss term $\mathcal{L}_{fc}$ (*functional consistency*) ensures that the prediction for the counterfactual matches the desired target $\bar{y}$, e.g., categorical cross entropy. As illustrated in Figure 3, the manifold consistency objective plays a central role in deep inversion-based CF generation. Even with a strong image prior (DIP in this example) and LSO, the generator produces out-of-distribution (OOD) CFs (with missing pixels) as guided by the semantic preservation term. Though the synthesized CF produces the desired class label with the classifier $\mathcal{F}$, the explanation is not interpretable. In contrast, including the $\mathcal{L}_{mc}$ term (using DEP explained next) leads to a meaningful counterfactual that automatically fills in the missing pixels. Note that, this example is different from prior-based *image inpainting* [16], where a known mask is used to alter the loss function to recover the missing pixels. In this work, we explore two different strategies to implement $\mathcal{L}_{mc}$:



Figure 3: *Need for manifold consistency.* Without explicitly constraining the CFs to lie close to the true manifold, deep inversion-based generators can produce OOD images (missing pixels) that satisfy *functional consistency*. In contrast, our approach is able to create a more faithful explanation by automatically filling in missing pixels.

**a. Direct Error Prediction (DEP)**. Recently, in [19], it was found that a loss predictor trained jointly with the classifier can be used to effectively detect distribution shifts and obtain accurate uncertainty estimates for a given sample. Assuming that, a classifier model $\mathcal{F}$ is trained to optimize the primary loss $\mathcal{L}_{pri} = \mathcal{L}_{CE}(\mathcal{F}(x), y)$, we construct an auxiliary loss predictor $\mathcal{G}$ to estimate the loss $s = \mathcal{G}(x) = \mathcal{L}_{pri}$. Similar to [18, 19], we utilize an auxiliary loss function $\mathcal{L}_{aux}(s, \hat{s})$ to train the parameters of $\mathcal{G}$. In particular, we adopt the contrastive loss which aims to preserve the ordering of samples based on their corresponding losses from $\mathcal{F}$. Let $s_i$ and $s_j$ denote the losses of samples $x_i$ and $x_j$, while the corresponding estimates from $\mathcal{G}$ are $\hat{s}_i$ and $\hat{s}_j$ respectively. Now,

$$\mathcal{L}_{aux} = \sum_{(i,j)} \max \left( 0, -\mathbb{I}(s_i, s_j).(\hat{s}_i - \hat{s}_j) + \gamma \right), \tag{6}$$

$$\text{where } \mathbb{I}(s_i, s_j) = \begin{cases} 1, & \text{if } s_i > s_j, \\ -1, & \text{otherwise.} \end{cases}$$

Here $\gamma$ is an optional margin parameter ($\gamma = 1$ for our implementation). The overall objective for the joint optimization of the classifier and loss predictor is given by

$$\mathcal{L}_{total} = \beta_1 \mathcal{L}_{pri} + \beta_2 \mathcal{L}_{aux}. \tag{7}$$

After the models are trained, we implement the manifold consistency term $\mathcal{L}_{mc}$ directly based on the loss predictor, since the losses indicate regimes where the model fails to make an accurate prediction:

$$\mathcal{L}_{mc} = \|\mathcal{G}(\bar{x}) - s^*\|_1, \tag{8}$$

where $s^*$ denotes the target loss to be achieved. In our experiments, we set $s^*$ as the median error estimate from DEP on a held-out validation set.

**b. Deterministic Uncertainty Quantification (DUQ)**. The recently proposed DUQ [20] method is based on Radial Basis Function (RBF) [28] networks and has been showed to be highly effective at OOD detection. In its formulation, a model is comprised of a deep feature extractor $\mathcal{F}$, an exponential

kernel function along with a set of prototypical feature vectors (or *centroids*) for each class. DUQ is trained by optimizing the kernel distance between the features from $\mathcal{F}$ and the class-specific centroids and using a moving average process to update the centroids. Once DUQ is trained, the uncertainty can be measured as the distance between the model output and the closest centroid. Details regarding DUQ training can be found in the appendix. In this work, we implement $\mathcal{L}_{mc}$ using a margin-based loss that maximizes the kernel similarity of the synthesized CF with the centroid of the target class, relative to the source class. Denoting the kernel similarity for a CF $\bar{\text{x}}$ with the centroid for class y as $K(\mathcal{F}(\bar{\text{x}}), \phi(\text{y}))$, where $\phi(\text{y})$ corresponds the pre-computed centroid for class $y$, we define:

$$\mathcal{L}_{mc} = \max\left( K(\mathcal{F}(\bar{\text{x}}), \phi(\text{y})) - K(\mathcal{F}(\bar{\text{x}}), \phi(\bar{\text{y}})) + \tau, 0 \right), \tag{9}$$

which indicates that kernel similarity w.r.t. the target class $\bar{\text{y}}$ should be greater than that with the source class y at least by the margin $\tau$ (set to $0.5$ in our experiments).

### 3.4 Progressive Optimization

CF generation is a highly under-constrained problem, that even with a strong image prior and the proposed manifold consistency constraint, it can easily converge to trivial solutions, i.e., irrelevant image manipulations. For example, one might expect to introduce large discernible changes by reducing the penalty $\lambda_1$ for semantics preservation. However, given the large solution space (defined by the number of parameters in the DIP/INR generator $f_\theta$), this often leads to unrealistic images. To circumvent this, we propose to adopt a progressive optimization strategy that gradually increases the number of layers in $f_\theta$ to be optimized and steadily relaxing the penalty $\lambda_1$ (by factor $\kappa$) to allow for larger, yet interpretable, changes. More specifically, denoting the number of layers in $f_\theta$ by $L$, in each iteration we train the parameters of the first $i$ layers ($i$ is incremented by 1 in the subsequent iteration) while keeping the parameters of the remaining $L - i$ layers at their initial state (details on how the layers are chosen for DIP and INR based generators can be found in the appendix). A similar strategy has been shown to be effective for ill-posed restoration tasks using large-scale generative models such as Style-GAN [29, 30]. An outline of this progressive optimization process is provided in the appendix. We find that such a progressive optimization leads to significantly better quality solutions allowing meaningful traversal from one class to another.

### 3.5 Evaluating Quality of CF Explanations using Classifier Discrepancy

A desired property in query-based explainers is that the synthesized changes are both interpretable and representative of the *target* class (*e.g., smiling*). To systematically evaluate the latter property, we propose the following synthetic experiment: Given a binary classifier $\mathcal{F}$ and training images, i.e., $X_0$, belonging to Class 0, we use our CF generator to synthesize examples for Class 1, i.e., $\bar{X}_1$ (using Class 0 images as input), and finally build a secondary classifier $\mathcal{F}^c$ using $[X_0, \bar{X}_1]$. The quality of the counterfactuals can thus be measured using the gap between the performance of $\mathcal{F}$ and $\mathcal{F}^c$ on a common test set. We refer to this score as *classifier discrepancy* (CD).

## 4 Experiment Setup

**Datasets.** (i) *CelebA Faces* [31]: This dataset contains 202,599 images along with a wide-range of attributes. For our experiments, we consider 3 different attributes, namely *smiling*, *bald* and *young*. Note, we train a classifier for predicting each of the attributes independently. We report the results for *bald* and *young* attributes in the appendix; (ii) *ISIC 2018 Skin Lesion Dataset* [32]: This lesion diagnosis challenge dataset contains a total of $10,015$ dermoscopic lesion images from the HAM10000 database [33]. Each image is associated with one out of 7 disease states: Melanoma (MEL), Melanocytic nevus (MN), Basal cell carcinoma (BCC), Actinic keratosis (AK), Benign keratosis (BK), Dermatofibroma (DF) and Vascular lesion (VASC). Note, in all cases, we used a stratified $90 - 10$ data split to train the classifiers.

**Classifier Design.** For all experiments, we resized the images to size $96 \times 96$ and used the standard ResNet-18 architecture [34] to train the classifier model with the Adam optimizer [35], batch size $128$, learning rate $1e - 4$ and momentum $0.9$. For the DEP implementation (Section 3.3), we performed average pooling on feature maps from each of the residual blocks in ResNet-18, and applied a linear layer of 128 units with ReLU activation. The hyper-parameters in (7) were set at $\beta_1 = 1.0$ and

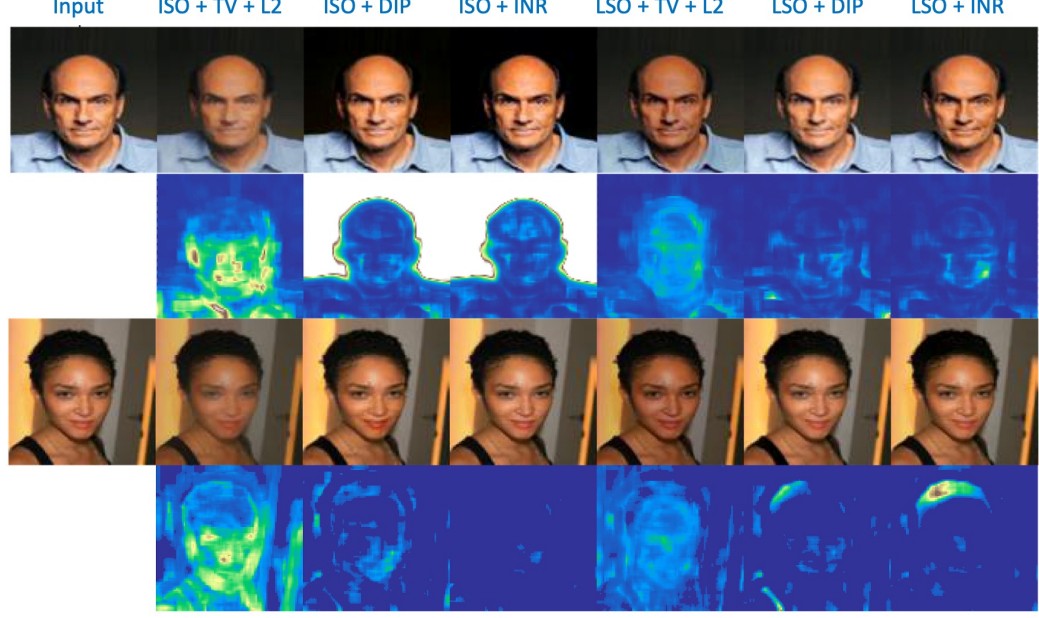

| Input | ISO + TV + L2 | ISO + DIP | ISO + INR | LSO + TV + L2 | LSO + DIP | LSO + INR |

Figure 4: *ISO vs LSO with different choices of priors*. Though none of the image priors inherently lead to discernible changes that reflect the properties of the target *smiling* class, we find that LSO with strong priors produces higher quality images compared to ISO.

$\beta_2 = 0.5$. For the case of DUQ, we set both the length scale parameter and the gradient penalty to 0.5.

**Image Generator Design.** For generator design, the deep image prior used the standard U-Net architecture and input noise images drawn from the uniform distribution $\mathcal{U}[-1, 1]$. For INR, we chose 256 random sinusoids with frequencies $b_i$ drawn from a Gaussian distribution with mean 0 and variance 100 to compute the Fourier mapping for the input coordinates.

## 5 Findings

**LSO Offers Better Feature Manipulation Control over ISO with Different Priors**. An important design component in DISC is how we compute the semantic discrepancy between query $x$ and CF $\bar{x}$ - either in the pixel space using ISO or in the latent space of the classifier using LSO. We perform a comparative analysis of their behavior in CF generation using CelebA faces, in particular when manipulating an image from the *non-smiling* class to be classified as *smiling*, by varying the choice of image priors. As observed in Figure 4, though LSO produces higher quality images compared to ISO when using a weak image prior (TV + $\ell_2$), that quality gap vanishes with the use of a stronger prior, e.g., DIP. However, in terms of producing discernible changes that reflect the properties of the *smiling* class, neither approach is sufficient. In particular, ISO shows a higher risk of making minimal, irrelevant modifications (refer to difference images $|x - \bar{x}|$ in Figure 4) that drive the prediction to a desired label and hence, similar to [13], we recommend the use of LSO but with stronger image priors to design CF generators.

**Manifold Consistency is Essential to Produce Meaningful Explanations**. As showed in the previous experiment, deep model inversion does not produce discernible (and interpretable) image changes when applied for CF generation. In this context, we explore the impact of enforcing manifold consistency in DISC. In particular, we compare the following LSO-based CF generation implementations (with INR prior): (i) no manifold consistency; (ii) DUQ-based $\mathcal{L}_{mc}$ from (9); and (iii) DEP-based $\mathcal{L}_{mc}$ from (8). From the results in Figure 5, we find that the proposed DEP objective significantly improves over the standard LSO (with no $\mathcal{L}_{mc}$) by introducing appropriate pixel manipulations near the mouth and cheeks in all examples and clearly represents the underlying semantics of the *smiling* class. In contrast, the RBF network-based DUQ performs very similar to the *LSO + INR* baseline and

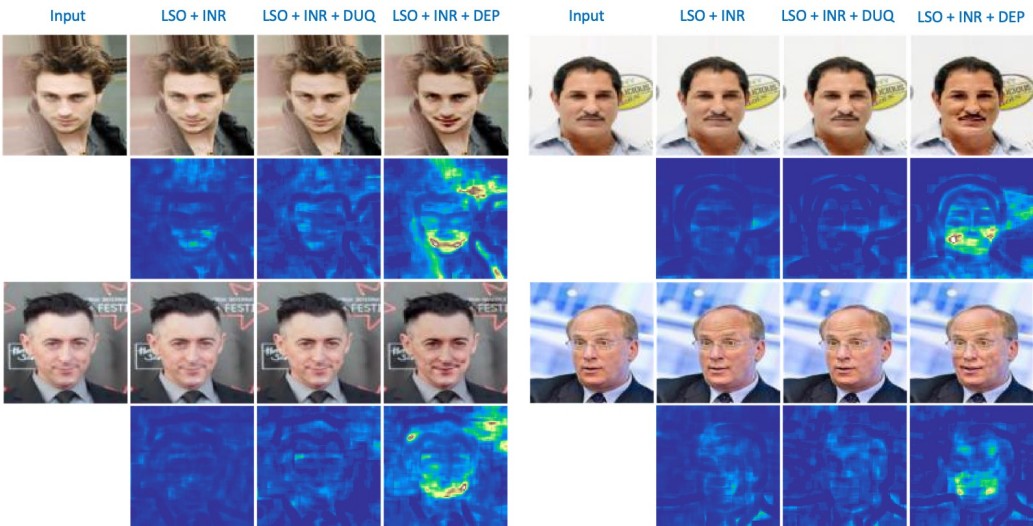

Figure 5: *Importance of manifold consistency*. The proposed DEP objective significantly improves over the standard LSO (with no $\mathcal{L}_{mc}$) by introducing appropriate pixel manipulations near the mouth and cheeks in all examples. In contrast, we find that DUQ-based consistency is insufficient to emphasize the semantics of the *smiling* class as seen in the difference images ($|\mathrm{x} - \bar{\mathrm{x}}|$).

Table 1: Evaluating the quality of the synthesized CFs on CelebA faces and ISIC2018 skin lesion datasets. The MSE and concentration metrics for the CelebA dataset were obtained using CFs synthesized for 5000 images from the *non-smiling* class. On the other hand, for ISIC2018, we used 800 images from the *MEL* class and generated CFs for changing the prediction to *NEV*.

| Dataset | Metric | Method | | | | |
|---------|--------|--------------|-----------|-----------|--------------|--------------|
| | | LSO + TV + L2 | LSO + DIP | LSO + INR | LSO + DIP + DEP | LSO + INR + DEP |
| CelebA | MSE | $0.36 \pm 0.18$ | $\mathbf{0.09 \pm 0.05}$ | $0.11 \pm 0.07$ | $0.19 \pm 0.11$ | $0.22 \pm 0.07$ |
| | Conc. | $0.43 \pm 0.22$ | $0.29 \pm 0.16$ | $0.28 \pm 0.19$ | $0.26 \pm 0.15$ | $\mathbf{0.18 \pm 0.11}$ |
| | CD | $0.31 \pm 0.05$ | $0.23 \pm 0.03$ | $0.26 \pm 0.06$ | $0.15 \pm 0.04$ | $\mathbf{0.08 \pm 0.03}$ |
| ISIC | MSE | $0.43 \pm 0.17$ | $\mathbf{0.14 \pm 0.08}$ | $0.19 \pm 0.09$ | $0.23 \pm 0.14$ | $0.24 \pm 0.13$ |
| | Conc. | $0.36 \pm 0.16$ | $0.32 \pm 0.14$ | $0.29 \pm 0.13$ | $0.25 \pm 0.09$ | $\mathbf{0.22 \pm 0.07}$ |
| | CD | $0.32 \pm 0.13$ | $0.28 \pm 0.15$ | $0.25 \pm 0.12$ | $0.17 \pm 0.05$ | $\mathbf{0.11 \pm 0.01}$ |

this emphasizes the inability of the kernel similarity metric to detect mild distribution shifts in data, though it has been proven successful for detecting severely OOD samples.

**INR based Priors Produce Highly Concentrated Changes over DIP**. Although both DIP and INR are effective, a more rigorous comparison of those two image priors is required. For this purpose, we utilize two important evaluation metrics, namely: (i) ability to produce large discernible changes, measured using $\ell_2$ error between x and $\bar{\mathrm{x}}$ in the pixel-space; and (ii) ability to produce *concentrated* image manipulations [36], measured by thresholding ($< 0.05$) the difference image $|\mathrm{x} - \bar{\mathrm{x}}|$ and determining the area of the largest bounding box that contains all the non-zero values (between 0 and 1). Ideally, explanations that have a larger MSE and consistently lower concentration are more likely to produce easily interpretable, localized changes. The results in Table 1 are obtained by randomly choosing 5000 images from the *non-smiling* class and synthesizing the corresponding counterfactuals for *smiling*. Similarly, for the ISIC2018 dataset, we used 800 randomly chosen images from the *MEL* class and generated CFs for *NEV*. The naïve *LSO + TV + $\ell_2$* baseline produces counterfactuals with a significantly large MSE as well as concentration score, indicating that the CFs are uninterpretable and contain irrelevant perturbations all over the image. As expected, incorporating a stronger prior improves the concentration significantly, while also being highly conservative in terms of MSE, *i.e.,* non-discernible changes. Finally, enforcing manifold consistency using DEP achieves an optimal trade-off between the two metrics and produces meaningful CFs (Figures 6 and 7). In particular, INR based generators produce highly concentrated image manipulations.

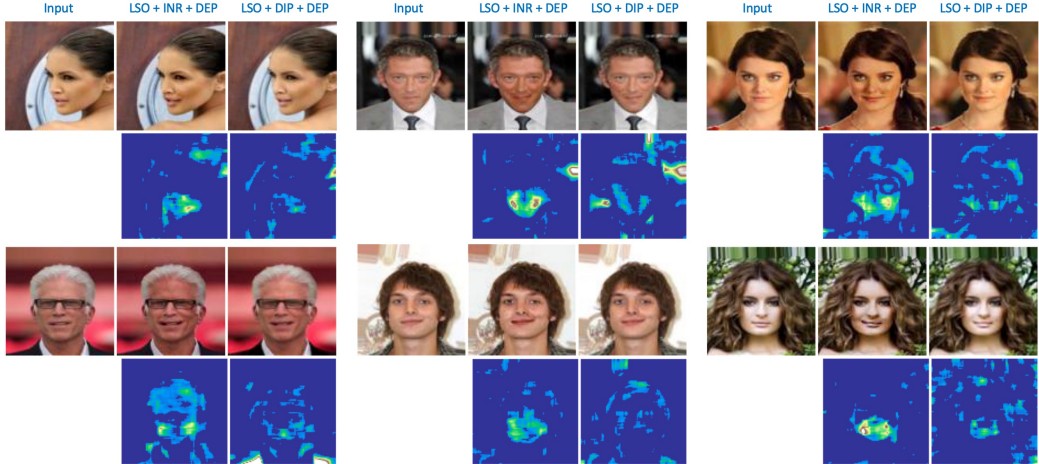

Figure 6: *Comparison between DIP and INR with DEP manifold consistency*. Although both DIP and INR are effective for LSO-based model inversion, we find that INR based generators produce highly concentrated and more apparent image manipulations.

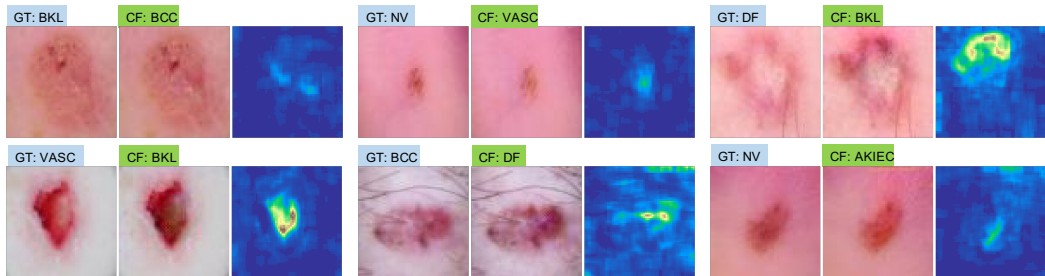

Figure 7: *Observations on CF synthesis for examples from ISIC 2018 dataset*. We find that, even in a multi-class problem, our approach is able to synthesize concentrated image changes, thus enabling us to introspect deep models with arbitrarily complex decision boundaries. Moreover, such perturbations are consistent with the ABCD (asymmetry, border, color and diameter) signatures adopted by clinicians for diagnosing lesions.

**LSO + INR + DEP Produces CFs with Low Classifier Discrepancy Score.** We now evaluate the quality of the counterfactuals using the proposed *classifier discrepancy* (CD) score. For this purpose, we consider a random subset of 5000 images each from *non-smiling* ($X_0$) and *smiling* ($X_1$) classes respectively. Following the strategy described in Section 3.5, we train the classifiers $\mathcal{F}$ and $\mathcal{F}^c$, and measure the CD score as the difference in test accuracies, $Acc.(\mathcal{F}, X^{test}) - Acc.(\mathcal{F}^c, X^{test})$. Similarly, we repeat a simple evaluation for ISIC data by using 800 images from class *MEL* as $X_0$ and images from class *NEV* as $X_1$. From Table 1, we find that, using manifold consistency along with a strong image prior produces significantly lower CD scores (0.08 with *LSO + INR + DEP* on CelebA), when compared with LSO without manifold consistency (0.26 with *LSO + INR*). In particular, using INR + DEP fairs the best and consistently produces highly meaningful counterfactuals.

**DISC Explanations are Robust Under Test-Time Distribution Shifts**. In several practical applications, we often encounter shifts between the train and test distributions which makes model deployment challenging. Hence, we evaluate the behavior of the proposed approach under unknown distribution shifts at test time. Note, we train the classifier on the clean CelebA faces dataset without introducing any corruptions. However, when we introduce corruptions at test-time, we find that (see Figure 8), our approach (*LSO + INR + DEP)* is able to robustly manipulate the appropriate regions in the query image. This can be attributed both to the ability of DEP to reflect challenging distribution shifts with higher error estimates [19] and our progressive optimization strategy to induce larger yet semantically meaningful changes (i.e., inherent noise clean up).

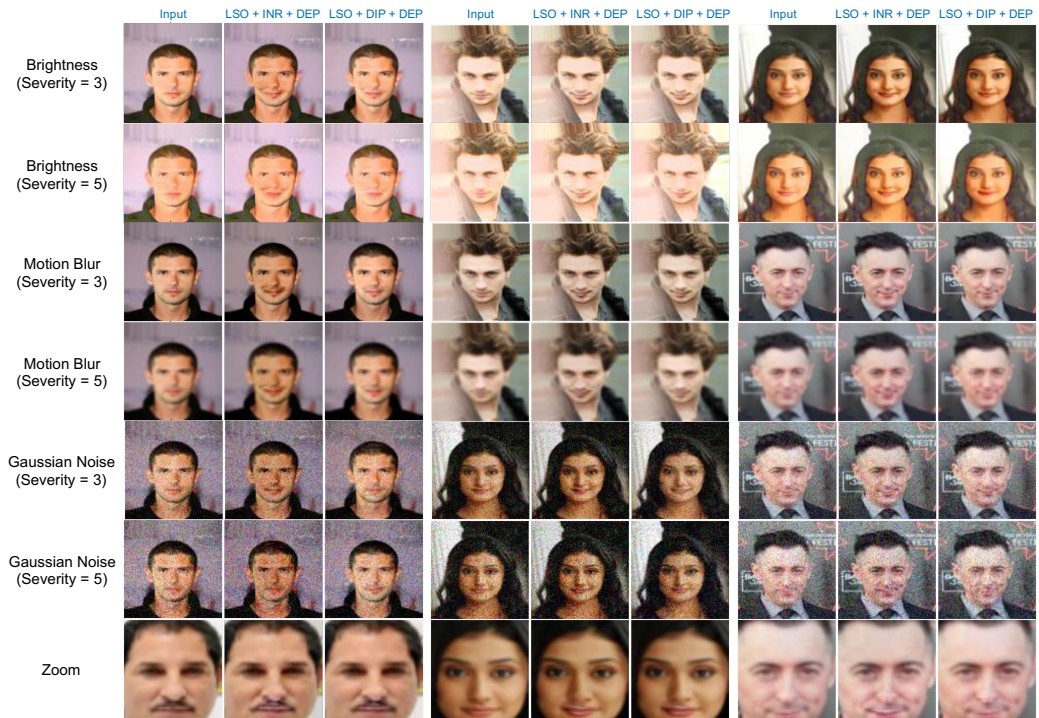

Figure 8: *DISC explanations are robust under test-time corruptions*. We find that even under unknown test-time corruptions, our approach robustly manipulates the appropriate regions in the query image (e.g., mouth and cheeks for smiling). Such a behaviour can be attributed both to the ability of DEP to reflect challenging distribution shifts [19] and our progressive optimization.

## 6 Conclusions

In this paper, we presented DISC, a general approach to design counterfactual generators for any deep classifier, without requiring access to the training data or generative models. We draw connections to the problem of deep model inversion and extend it to support counterfactual generation. DISC can learn a CF generator on-the-fly by leveraging different image priors and manifold consistency constraints, along with a progressive optimization strategy, to synthesize highly-plausible explanations. Future extensions to this work include exploring the use of multiple target attributes simultaneously in our optimization and applying this method to time-varying data.

## 7 Acknowledgements

This work was performed under the auspices of the U.S. Department of Energy by Lawrence Livermore National Laboratory under Contract DE-AC52-07NA27344.

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
