# Supplementary - Designing Counterfactual Generators using Deep Model Inversion

**Jayaraman J. Thiagarajan**
Lawrence Livermore National Laboratory
jjayaram@llnl.gov

**Vivek Narayanaswamy**
Arizona State University
vnaray29@asu.edu

**Deepta Rajan**
IBM Research AI
r.deepta@gmail.com

**Jason Liang**
Stanford University
jialiang@stanford.edu

**Akshay Chaudhari**
Stanford University
akshaysc@stanford.edu

**Andreas Spanias**
Arizona State University
spanias@asu.edu

## Appendix A - Design of DIP/INR Image Generators

**DIP architecture**. Following Ulyanov *et al.*, we designed the Deep Image Prior (DIP) using the standard U-Net architecture. In particular, the U-Net consists of $4$ downstream and upstream blocks with skip connections between each downstream and upstream block. Each downstream block $i$ contains two Conv2D layers with filters $\{16.2^{i-1}, 16.2^{i-1}\}$. Correspondingly, the upstream block $j$, comprises of a *bi-linear* up-sampling step followed by two Conv2D layers with filters sizes $\{16.2^{j-1}, 16.2^{j-1}\}$. We employ a kernel size of 3 for all the Conv2$D$ and incorporate BatchNorm2D for all layers. In all our experiments, input noise images were drawn from the uniform distribution $\mathcal{U}[-1, 1] \in \mathbb{R}^{3 \times 96 \times 96}$.

**INR architecture**. The coordinate-based INR model was constructed using $5$ fully connected layers with $512$ hidden units and a final layer with $3$ outputs corresponding to the RGB channels. In our implementation, we choose $256$ random sinusoids (basis) with frequencies $b_i$ drawn from a Gaussian distribution with mean $0$ and variance $100$ to compute the Fourier mapping for the input coordinates.

## Appendix B - DUQ Implementation

We adopted the existing code from Amersfoort *et al.* to train the DUQ models. In particular, we utilized a ResNet-18 as the feature extractor to generate $d = 512$ dimensional feature vectors for every image. Correspondingly, the class-specific centroids were also of dimension $D = 512$. To improve stability in model training, we used a gradient penalty and length scale parameter of $0.5$ and smoothing factor $\gamma = 0.999$ for the moving average process while updating the centroids.

## Appendix C - Progressive Optimization

CF generation is a highly under-constrained problem, and even with a strong image prior such as DIP/INR and the proposed manifold consistency, it can still be challenging to avoid trivial solutions. One can expect to introduce large discernible changes by reducing the penalty $\lambda_1$ for semantics preservation. However, given the large solution space, this often leads to unrealistic images. To overcome this, we propose to adopt a progressive optimization strategy that gradually increases the number of layers in the DIP/INR generator to be optimized and steadily relax the penalty $\lambda_1$ (by factor $\kappa$) to allow for larger, yet interpretable changes. Algorithms 1 and 2 provide the details of the proposed progressive optimization.

---

**Algorithm 1** Progressive Optimization with DIP

---

1: **Input**: Query Image x, Target label $\bar{y}$, number of rounds $R$, number of epochs $N$;
  UNet-based DIP $f$ with $L$ downstream ($d$) and upstream ($u$) blocks with parameters
  $\theta = \{\theta_d\} \bigcup \{\theta_u\}$;
  Penalties $\lambda_1, \lambda_2, \lambda_3$, factor $\kappa$, learning rate $\eta$, Pre-trained classifier $\mathcal{F}$;
2: **Output**: Counterfactual explanation.
3: **Initialize**: Input noise image $z \in \mathcal{U}[-1, 1]$, number of layers to optimize $l$;
4: **for** $r$ in 1 to $R$ **do**
5:   $\theta_r = \{\theta_d^{1:l}\} \bigcup \{\theta_u^{1:l}\}$;
6:   **for** $n$ in 1 to $N$ **do**
7:     Generate $\bar{x} = f_\theta(z)$;
8:     Compute loss $\mathcal{L}(x, \bar{x}, \mathcal{F})$ using Eqn (5);
9:     $\theta_r \leftarrow \theta_r - \eta \nabla_{\theta_r} \mathcal{L}$;
10:   **end for**
11:   $l \leftarrow \min(l+1, L)$;
12:   $\lambda_1 \leftarrow \kappa * \lambda_1$;
13: **end for**
14: **return**: Counterfactual Image $\bar{x} = f_\theta(\mathbf{z})$.

---

---

**Algorithm 2** Progressive Optimization with INR

---

1: **Input**: Query Image x, Target label $\bar{y}$, number of rounds $R$, number of epochs $N$;
  INR-based generator $f$ with $L$ layers and parameters $\theta$;
  Penalties $\lambda_1, \lambda_2, \lambda_3$, factor $\kappa$, learning rate $\eta$, Pre-trained classifier $\mathcal{F}$;
2: **Output**: Counterfactual explanation $\bar{x}$.
3: **Initialize**: Input coordinates v, number of layers to optimize $l$;
4: **for** $r$ in 1 to $R$ **do**
5:   $\theta_r = \{\theta^{1:l}\}$;
6:   **for** $n$ in 1 to $N$ **do**
7:     Computer Fourier mapping $z = B(v)$;
8:     Generate $\bar{x} = f_\theta(z)$;
9:     Compute loss $\mathcal{L}(x, \bar{x}, \mathcal{F})$ using Eqn (5);
10:     $\theta_r \leftarrow \theta_r - \eta \nabla_{\theta_r} \mathcal{L}$;
11:   **end for**
12:   $l \leftarrow \min(l+1, L)$;
13:   $\lambda_1 \leftarrow \kappa * \lambda_1$;
14: **end for**
15: **return**: Counterfactual Image $\bar{x}$.

---

## Appendix D - Results for CelebA - Baldness Attribute

For this experiment, we used the CelebA faces dataset and considered the *baldness* attribute. Following the experiment setup that we used for the *smiling* attribute, we generate counterfactuals that change the prediction from *not bald* to the *bald* class. As showed in Figure 1a, the results from DISC (with INR generator and DEP-based manifold consistency) leads to highly meaningful explanations, wherein the hairline is appropriately modified to increase the likelihood of the *bald* class.

## Appendix E - Results for CelebA - Age Attribute

In this experiment, we use the *age* attribute for the CelebA faces and employ DISC to systematically manipulate the input image (*young*) and ensure that it is classified as *old*. Results from Figure 1b show that DISC prefers to manipulate the region around the eyes in order to increase the likelihood for the *old* class.

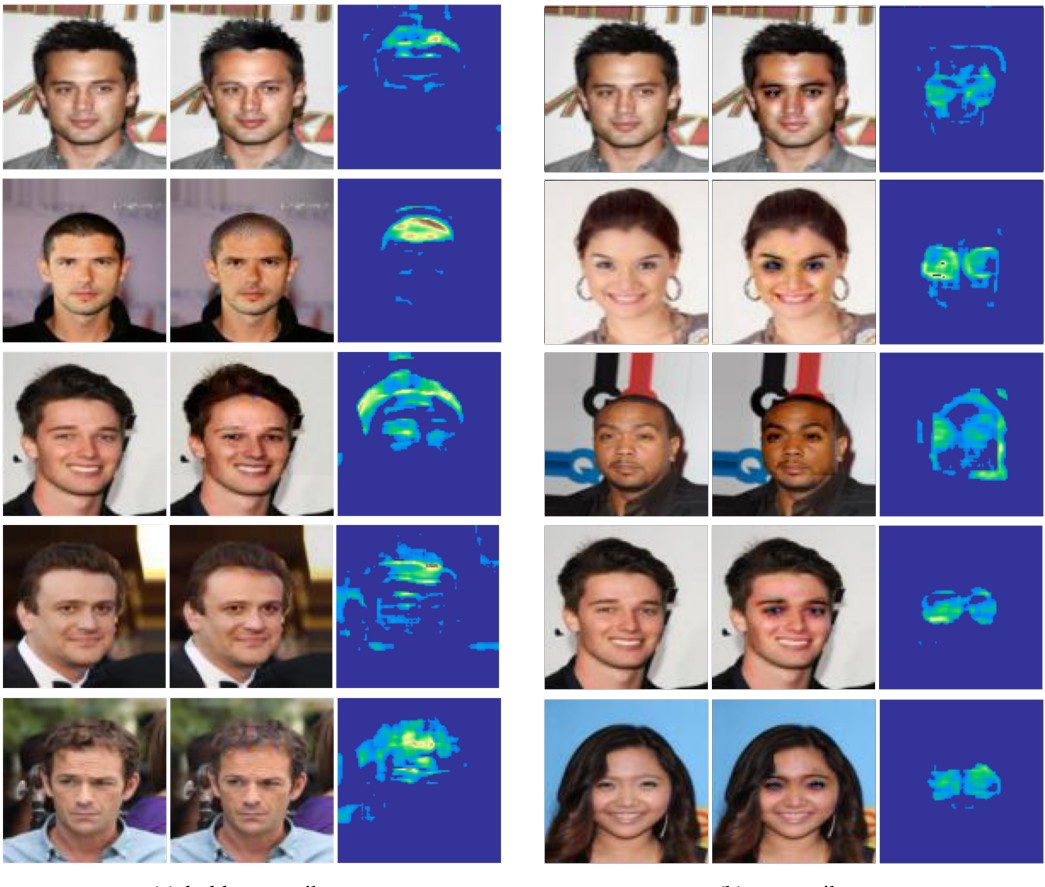

(a) *baldness* attribute   (b) *age* attribute

Figure 1: Examples of counterfactuals generated for the *baldness* and *age* attributes using DISC. Here the inputs (left) were from the *not bald* or *young* class. Using the implementation of DISC with INR image generator and DEP-based manifold consistency, we obtain the CFs (middle) for the *bald* and the *old* classes respectively. As illustrated in the difference image (right), we clearly notice that DISC introduces meaningful perturbations to the image.

## Appendix F - Additional Results for ISIC2018

In the experiments reported in the main paper, we used the ISIC2018 skin lesion classification dataset and demonstrated how DISC can be used to traverse complex decision boundaries. In this section, we show additional examples from that experiment to further emphasize the effectiveness of DISC (INR generator + DEP-based manifold consistency). Figure 2 illustrates transition between Vascular (VASC) and Benign keratosis lesions (BKL), as well as, between Basal cell carcinoma (BCC) and Dermatofibroma (DF). In the first case, Benign keratosis lesions are known to be characterized by significant differences in both color and intensity when compared to vascular lesions. Consequently, DISC produces highly concentrated pixel manipulations on the actual lesion regions of each image. On the other hand, changing the prediction from Dermatofibroma to Basal cell carcinoma requires more global changes pertinent to both border and asymmetry properties of the lesions.

## Appendix G - Comparison with Generative Model based Counterfactual (CF) Generation Methods

Though our focus was on test-time CF generation without assuming access to generative models or training data, we generated results for a GAN-based baseline. As expected, the image quality is better with pre-trained generative models (as indicated by lower FID, and higher precision/recall

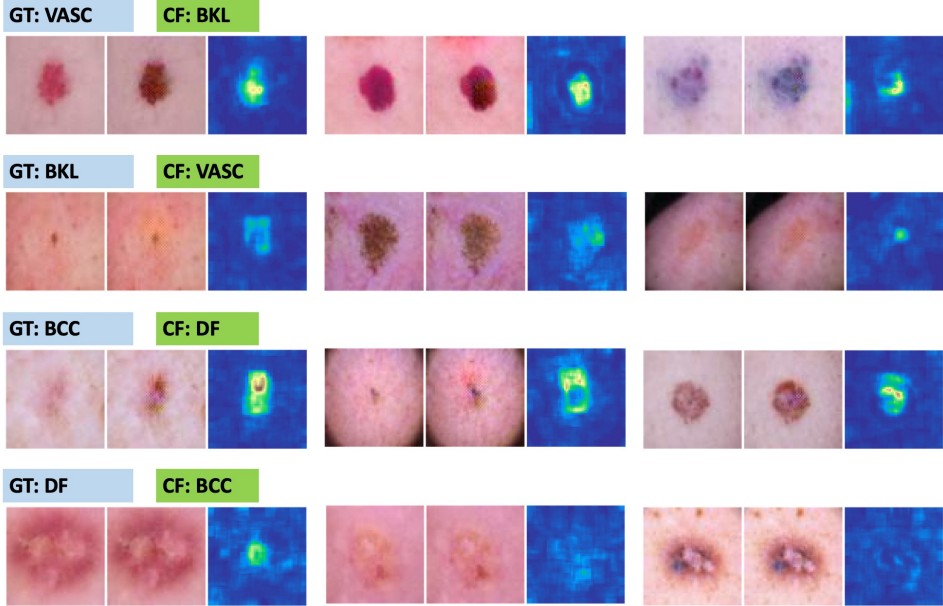

Figure 2: Additional examples from the ISIC2018 skin lesion classification dataset.

scores). However, in terms of evaluation metrics such as the proposed classifier discrepancy (CD), it is not significantly different than our proposed approach (DISC). This clearly demonstrates the ability of DISC to effectively manipulate the most relevant image features and produce representative counterfactuals for the target class. The results for this comparison are for the case of CelebA dataset (smiling attribute) is provided in Table 1.

Table 1: Comparing GAN-based CF generation against DISC. The results for this comparison are for the case of the CelebA dataset (non smiling –> smiling). We provide the metrics averaged over 5000 realizations.

| Method | FID | Precision | Recall | Classifier Discrepancy |
|---|---|---|---|---|
| GAN Based (Sauer & Geiger, 2021) | **69.4** | **0.71** | **0.27** | **0.07±0.04** |
| DISC | 81.7 | 0.66 | 0.19 | **0.08±0.03** |

## Appendix H - Identifying Changes Relevant for Classification

Table 2: Comparing saliency maps of 50 randomly chosen smiling images from DISC and GradCAM by zero-masking top 15% of influential features by computing $\Delta$ log-odds.

| | Grad-CAM | DISC |
|---|---|---|
| $\Delta$ log-odds | $10.4 \pm 3.3$ | $\mathbf{11.1 \pm 3.7}$ |

Though DISC is designed for generating counterfactuals for different user-specified hypotheses on the predictions, it can also be re-purposed for obtaining saliency maps. We compute the saliency map for a target class as follows: $|\mathcal{C}(x, y = 1) - \mathcal{C}(x, y = 0)|$, where $x$ is the query image, $\mathcal{C}(x)$ is the counterfactual and $y$ is the target class. We find that saliency maps from DISC are very comparable to standard approaches such as GradCAM in terms of the $\Delta$ log-odds metric. Formally, $\Delta$ log-odds = log-odds($\mathcal{F}(x)$) - log-odds($\mathcal{F}(x_{masked})$), where log-odds(p) = log(p/1-p) and $\mathcal{F}$ is the pre-trained classifier. For example, DISC saliency maps for the CelebA smiling–>not smiling classifier picks the regions near the mouth and cheeks as the influential features. In order to understand the quality of the DISC based saliency maps with GradCAM, we use 50 randomly chosen smiling images from the CelebA dataset and zero mask the top 15% of the influential features relevant for classification to

compute the $\Delta$ log-odds metric. It can be observed that (Table 2), DISC infact produces meaningful changes to the salient regions of images that are highly reflective of the underlying target class.