# OpenReview forum: "Designing Counterfactual Generators using Deep Model Inversion"
_NeurIPS.cc/2021/Conference — NeurIPS 2021 Poster_

### Official Review · Reviewer_za8w · 2021-07-16

**Rating:** 7
**Confidence:** 3

**Summary:**

**(motivation)** Many approaches exist to generate counterfactual images, but what should be done when no training data is available? Current approaches lead to adversarial-looking approaches, which are not satisfying enough as an explanation.

**(approach)** The authors use Deep Inversion and a novel objective that ensures the data stays in distribution and propose a method to evaluate the quality of counterfactuals.

**(experiments)** The authors run experiments on Celebi and ISIC.



**Main Review:**

**Strengths**

- The work is well-motivated and thoroughly describes the problems that counterfactual generation faces.
- The method is evaluated on two image datasets.
- The approach is novel.
- The work proposes a new metric for evaluating the counterfactuals, which might be less susceptible to bias.

## **Weaknesses**

- Since the authors propose a new metric, it makes sense to compare/discuss why the proposed approach is more appropriate than currently existing methods. However, a related work section about metrics for counterfactual evaluation is missing. It would strengthen the paper to present a comparison between the proposed methods and that of Dabkwoski & Gal (https://arxiv.org/abs/1705.07857), which is a close metric to what the authors propose.
- The authors do not evaluate against a baseline. While it is fine to exclude many methods for their reliance on training data, there do exist some methods without this limitation, e.g., Occlusion or Goyal et al. (https://arxiv.org/abs/1904.07451 ). This would make the evaluation more sound and strengthen it, as the existing methods probably perform worse.
- I can see only minor changes for some of the images, especially for the baldness attribute, if the person in the image has a lot of hair. The method does not seem to add wrinkles or similar structures that come with age. Generally, minor changes of attributes need more discussion in the paper. Why does this happen?
- It would be good to know if any images are showing only adversarial examples and, if so, how many.
- The checklist is not filled out correctly. Code is missing.

## **Missing Details / Points of Confusion**

- Line 141: explicitly introduce z as noise

## **Presentation Weaknesses**

- Figure descriptions (e.g., 3) are not detailed enough: How are the heatmaps computed?
- There is no bold in tables for best values, and an explanation of what optimal values of MSE and concentration would be is missing in the description of Table 1

## **Suggestions**

- Experiment with baldness and age for both bald <-> hairy and young <-> old
- Explain why the proposed measure for evaluation is better and compare it with another metric.

## **Preliminary Evaluation**



**Clarity:** the paper is clear in its motivation and explanation of the method and experiments.

**Originality:** the problem and approach of creating counterfactuals without access to the training data is novel.

**Significance:** the problem is significant, especially if training data is not available.

**Quality:** the paper is well-written but the formatting of the tables and figures could be improved. Furthermore, the missing comparison with other methods does affect the quality of the paper.



**tl;dr:** This paper proposes an interesting approach worth publishing, but the experimental part should be improved by adding comparisons with another metric and a baseline method. It also could benefit from a more in-depth discussion of drawbacks (what happens when bias is involved, the danger of still finding adversarial changes only).

# After Rebuttal

I thank the reviewer for their extensive answer. They address my points well and I raised my score to 7.

**Time Spent Reviewing:**

3

---

> ### Author Response · Authors · 2021-08-10
> **Response to Reviewer za8w**
>
> We thank you for your constructive suggestions and comments. Here are our responses to your concerns and the changes that we will make to the manuscript.
>
> > **Importance of the Proposed Metric:**
>
> Conventionally, counterfactual reasoning methods use a wide-range of metrics that quantify the image quality, proximity to the true data manifold, sparsity of the explanation, concentration of the saliency map etc. In our evaluation, we followed standard practice and measured the amount of discernible change in the pixel space and its trade-off with the concentration metric. While these metrics are effective at evaluating single instances of counterfactual explanations, they cannot provide a holistic evaluation with respect to the entire dataset. To this end, we introduced the classifier discrepancy (CD) metric that more rigorously tests the ability of counterfactuals to describe the semantics of a target class. A lower value of CD (evaluated using the same test data) indicates that a classifier trained using only the counterfactuals as examples for the target class matches the performance of the original classifier and hence provides a comprehensive characterization of our approach. It is important to note that the metric is not specific to our method and can be used to evaluate any CF generator.
>
> > **Comparison with Dabkowski & Gal and Extension of Related Work:**
>
> The saliency metric proposed in Dabkowski & Gal involves determining the smallest crop in an image that can predict the target class. Similarly, we have adopted a concentration metric (Table 1 in the paper) that measures the area of the largest bounding box in the difference image (between the original image and the synthesized CF) that contains non-zero pixel values. A smaller value of concentration implies that the perturbations introduced are more local. To clarify this, we will extend our related work by including a section on the current metrics utilized to evaluate CFs in a potential final version of the paper.
>
> > **Other baselines / Comparison with Goyal et.al:**
>
> We thank the reviewer for suggesting the paper by Goyal et al. as a baseline for our approach. However, the approach proposed in their work also requires explicit knowledge about the underlying data manifold. In particular, a set of distractor images are required to introduce minimum edits in a given image, while DISC assumes no access to the training data and introduces plausible perturbations to the query image using only a pre-trained classifier and a test time image generator. Following your suggestion and the comments from other reviewers as well, we will definitely include a GAN-based baseline and also report image quality metrics (FID score and proximity to the data manifold) to demonstrate how our test-time approach compares. Here are some initial results (same as the table included in our response to Reviewer txfe)
>
> _Table A. Comparing GAN-based CF generation against DISC. CelebA Not Smiling -> Smiling (Results from 5000 synthesized images)_
>
> |Method&nbsp;|FID &nbsp;|Precision&nbsp;&nbsp;|Recall&nbsp;&nbsp;|Classifier Discrepancy|
> |:-:|:-:|:-:|:-:|:-:|
> |GAN-based &nbsp;|**69.4**|**0.71**|**0.27**|**0.07 $\pm$ 0.04**|
> |DISC&nbsp;|81.7&nbsp;|0.66|0.19|**0.08 $\pm$ 0.03**|
>
> > **Cause of minor changes in certain attributes:**
>
> This is a very important question. The degree of change in attributes depends on the properties of the classifier and the manifold consistency term used in our optimization. While our proposed progressive optimization strategy helps produce larger discernible changes when compared to single-shot optimization, for some attributes (e.g., baldness), DISC can introduce varying degrees of changes for different query images. This behavior is due to the observation that even with those small manipulations, the uncertainty-based manifold consistency term is minimized and hence is not able to provide any reward for making additional changes. Hence, in a test-time only optimization as ours, one potential way to introduce more drastic changes is to improve our confidence estimators (e.g., DEP in our case) that provide high confidences only for samples close to the prototypical samples (e.g., fully bald faces) and slightly lower confidence as we move away from the prototypes. However, we note that our DEP estimator is generally very effective, the changes are always concentrated in appropriate image regions and the resulting CFs lead to lower classifier discrepancy scores.
>
> > **Adversarial Examples produced by DISC:**
>
> When using the manifold consistency, we did not generally find adversarial corruptions in our explanations. On the other hand, the unconstrained version without the manifold consistency term often produces adversarial corruptions. From our experiments on the two datasets and different attributes, we find that DISC produces meaningful changes with high concentration values. In particular, we did a post-hoc analysis of the explanations from our skin lesion detection experiments and found that compared to all other variants, LSO + INR + DEP consistently produces changes to the lesion-specific regions and not to irrelevant background pixels. However, like any counterfactual reasoning method, DISC can also cause changes to confounding attributes that arise from the data bias. For example, even if the majority of the smiling faces in the training data are females, smiling counterfactuals for a non-smiling male face can introduce unexpected changes to some other attributes (like eye regions). Since the main paper does not have space for this experiment, we will include this to the final version of the supplementary material.
>
> > **Additional results:**
>
> In all datasets, we have completed the experiments for counterfactuals with the opposite hypothesis (bald -> not bald, smiling -> not smiling etc.) and could not include due to space constraints. The results are comparable to the cases reported in the paper, in terms of the classifier discrepancy and concentration metrics. Furthermore, we find that DISC is highly effective at identity preservation of a given query image, i.e., $x \rightarrow \mathcal{C}(x; y=1) \rightarrow \mathcal{C}(\mathcal{C}(x; y=1); y=0) \rightarrow x$. In other words, if we take a non-smiling query image, first generate a counterfactual $\mathcal{C}(x)$ for smiling and then generate the second counterfactual $\mathcal{C}(\mathcal{C}(x))$ for non-smiling, we are able to accurately recover the original query x. Here, we report the identity preservation error for 50 randomly chosen images from the not smiling class measured as: $\|\mathcal{C}(\mathcal{C}(x)) - x\|_1/255$.
>
> _Table F. DISC can preserve the identity of the query image,_ $\mathcal{C}(\mathcal{C}(x)) = x$
>
> |&nbsp;|Identity Preservation |
> |:-:|:-:|
> |LSO + INR + DEP|0.01 $\pm$ 0.008|
>
> > **Optimal values of MSE and Concentration metrics:**
>
> There is generally a trade-off between these metrics while evaluating the quality of CFs. A meaningful CF of a query image needs to contain larger yet discernible changes with concentrated image manipulations. CFs that have a larger MSE and consistently lower concentration are more likely to produce easily interpretable, localized changes while CFs that have a larger MSE and concentration metric are more likely to cause irrelevant pixel perturbations.
>
> > **Code Availability:**
>
> We will publicly release our code and include the link in a potential final version of the paper.
>
> > **Notations:**
> + In line 141, noise $\mathrm{z}$ is of the same dimensions as that of the query drawn from a uniform distribution $\mathcal{U}(-1,1)$.
> + In Fig. 6, the heatmaps are essentially the absolute differences between the true image and the CF.  We will update the figure captions to clarify this.
> + We will better format the tables and highlight the best scores in the potential final version of the paper.

---

### Official Review · Reviewer_hrJB · 2021-07-16

**Rating:** 7
**Confidence:** 4

**Summary:**

In this paper authors propose to generate counterfactuals from a query image and a given trained deep classifier. One is interested to find out a semantically reasonable alternative explanation, where image would be classified from class y to y'. This is all performed without the help of an explicit generator (such as GAN).

**Limitations And Societal Impact:**

Yes

**Main Review:**

If we think about adversarial examples, then yes changing the input with a minimal adversarial noise, we can obtain classification result to any target class. But such an "counterfactual" would not be of real use for practitioners as the noise is hand-crafted to mess  up the classification. Semantic and manifold constraints are needed to generate an image that has meaning.

Even though authors say that they do not use generator models, but in deep image priors, decoder model is included and I am thinking that constraining the search using the generator model is a pretty crucial feature. Similar work was done in activation maximization paper (that authors cite), where latent code of autoencoder is manipulated and the decoder generates the "counterfactual".

I would like to see  a bit more discussion in this paper about the meaning and usefulness of these kind of generated examples. Is the goal to generate good smiling images from non smiling ones, or is it to debug the deep model, where darkened eyes can lead to classifier to decide that image is old vs actually being young. Defining the final goal exactly can help then to conduct experiments that will ultimately find out whether the goal was reached or not. For example if the goal is to debug then, then it is quite important to know how the classifier was fooled by the new image. Some numerical measurement is then essential to show.

In the example CFs it would be interesting to see the strength of activation of the target class. Such as does semantically meaningful generation result in an image that actually is not strongly classified as target. Even better would be to quantify the uncertainty of classification, such as using BNN type schemes.

Ultimately, I see that authors have dug into deep learning toolbox to build the present system. What I liked about it is that it gives the idea that all of these options were tried out and we have thus arrived at an conclusion. But at the same time each particular block was known in advance, so technical novelty is in the combination.

Other comments:
M(x) in Eq. (1) is just one point? Maybe input to M(.) needs to be the whole training set? Or maybe the meaning is a local data manifold, centered on query point x?

What is R(x) in  Eq. (2) ? Obviously it needs to be a scalar valued function.  A more mathematical definition at this point is needed.

3.2.b should be explained with precision. Now it is hard to grasp how DIP would work, especially compared to a and b.

In 3.2.c.i what is b_1 ? In the next equation b is obviously scalar, but in this one it has to be 2-d vector.

What is s^* in (8)

In Section 3.5, classifier discrepancy measure should possibly also take into account the stochastic nature of training. This  would mean that multiple instances of  F^c are estimated from the same training set. Then the final CD measure is an interval instead of a point estimate.

In all Figures, the meaning of bluish sub-Figures is not explained. I am guessing that it is a pixel level difference image.

Figure 3, I personally do not see any of the images as smiling. To me this tells that this kind of empirical study should be reinforced with perceptual experiments.

Why L_mc is termed manifold consistency when you work on the score level only? I would imagine manifold consistency meaning that we take latent space of the images as make sure that we do not drift far from that latent image manifold. How do you reconcile this concept with the one used in your work?



**Time Spent Reviewing:**

6

---

> ### Author Response · Authors · 2021-08-10
> **Response to Reviewer hrJB**
>
> We are glad that you liked our paper and we sincerely thank you for your valuable comments and feedback:
>
> > **Goal of DISC:**
>
> DISC is a general approach for synthesizing plausible CFs by exploring the vicinity of a query image for a user-specified hypothesis. In this paper, we mainly focused on identifying and highlighting optimal design choices required to generate meaningful and realistic CFs on the fly (without needing access to training data or generative models). We believe DISC can be leveraged for model validation (saliency maps, progressive explanations etc.), identifying regimes of success and failure (e.g., shortcuts), inferring complex relationships between attributes (for example, in medical images, one might be interested in testing if age and disease severity are related) etc. Understanding the implications of DISC in these use-cases is definitely part of our future work. We will add this to a new discussion section in the paper (where we will also comment on the limitations of DISC).
>
> > **Relation to Activation Maximization:**
>
> Activation maximization aims to synthesize an arbitrary realization on the image manifold that maximizes the activation of a given neuron (in any layer). Note, in the paper we cite (Ulyanov et al.), this inverse problem is regularized only by the structure of the ‘encoder-decoder’ network (Deep Image Prior) while no latent code of the encoder is manipulated to generate the ‘counterfactual’ using the decoder. Although broadly related to our proposed approach, activation maximization is essentially a global explanation strategy that only points to key image features used by a classifier for making a decision. Moreover, without additional regularization, an arbitrary realization from activation maximization is not guaranteed to be on the data manifold and can even be an adversarial/out-of-distribution example that merely activates the target neuron. This is typically circumvented using approaches such as Deep Inversion (Yin et al.). Our proposed approach, DISC, on the other hand, is a local (query-level) explanation strategy that produces counterfactuals (CFs) for a given query image and a target class. Further, DISC adopts progressive optimization with a novel manifold closeness objective to regularize this challenging inverse problem.
>
> > **Prediction calibration of DISC counterfactuals:**
>
> When the classifier decision boundary is complex, it is possible that DISC optimization minimizes the function consistency term ($L_{fc}$) while making a larger error on the manifold consistency ($L_{mc}$). This corresponds to the scenario where DISC successfully produces a counterfactual for a given target class, but the predictor is not very confident, and hence not useful. We did find some cases in the multi-class classification experiment (ISIC 2018) where this happened due to the severe class imbalance (transitioning from a well-sampled to heavily under-sampled class). We will include relevant examples to the paper.  Note that, unreliability estimates from both the DEP and DUQ models (which are used to define $L_{mc}$) are directly correlated with the inherent model uncertainties and hence we do not require explicit uncertainty estimators (e.g., deep ensembles or Bayesian neural nets) to characterize prediction confidence.
>
> > **$L_{mc}$ is termed manifold consistency while we work on the score level:**
>
> While we operate at the score-level, the scores are inherently based on models (DEP or DUQ) that attempt to characterize the underlying data manifold. In fact, the estimates are directly indicative of how far a given sample is from the true manifold. Hence, we refer to this score as the manifold consistency objective. We will clarify this better in the paper.
>
> > **Measuring CD metric over different classifier instances / seeds:**
>
> The classifier discrepancy metrics reported in the paper (standard deviations in Table 1) were obtained using three independent trials of classifier training (with different random seeds). We will clarify that in the paper.
>
> > **Bluish sub-figures:**
>
> These are essentially the absolute differences between the true image and the CF.  We will update the figure captions in the paper to reflect this.
>
> > **Notations:**
>
> + In Eqn. 1, $\mathcal{M}(x)$ was a typo.  $\mathcal{M}(.)$ and not $\mathcal{M}(x)$ denotes the true image manifold
> + In Eqn. 2, $\mathcal{R}(x)$ is a general notation to denote the regularization strategy (Total Variation/ Deep Image Prior/ INR). We will clarify  the same in the paper.
> + We will improve the clarity of the notations of the Deep Image Prior and INR based regularizers in the paper.
> + In Section 3.2 c (i) in the design of the INR prior, $\{b_{i} |i=1, 2 … 255\}$ represents the frequency of random sinusoidal basis drawn from a Gaussian distribution with mean 0 and variance 100
> + We missed to include this in the paper and we will definitely update it in the final version. $s^*$ in Eqn. 8 denotes a target error estimate, we require our synthesized CF to produce. In our experiments we set $s^{*}$ as the median error estimate from DEP on a held-out validation set.
>
> > **Perceptual Experiments:**
>
> We are in complete agreement that the eventual utility of explainable AI methods is based on user feedback. Unfortunately, performing user studies was beyond the scope of this current paper and it is certainly part of our future work.

---

### Official Review · Reviewer_JzXC · 2021-07-17

**Rating:** 7
**Confidence:** 5

**Summary:**

Summary: The paper proposed a novel approach to achieve deep model inversion for a pre-trained deep classifier using the classifier alone, without having access to the training dataset or generative models. Given a query image, the proposed deep model inversion generates a counterfactual explanation that is realistic looking and produces the desired classification outcome.
Earlier work on image synthesis by inverting deep classifiers either creates unrealistic images or creates perturbation of query image that successfully changes the classification decision but resulting pixel-level manipulations are not interpretable/meaningful (similar to an adversarial attack).
The proposed approach has four essential components:
1.	Counterfactual image is synthesized using an image prior. The authors experimented with an un-trained UNET-based model as a deep image prior (DIP) and Fourier mapping with SIREN activation as implicit neural representation (INR) for mapping noise to the counterfactual image.
2.	To ensure high similarity between counterfactual and query images, the authors used ISO to minimize discrepancies over the entire image and LSO to minimize differences between embeddings extracted from different layers of the deep classifier.
3.	To ensure that the generated counterfactual image is realistic and lies on the data manifold, the proposed model uses a manifold consistency (MC) loss. The MC-loss is quantified using an auxiliary loss predictor function in DEP or the radial basis kernel similarity in DUQ. Both methods provide ways to capture epistemic uncertainty due to out-of-distribution (OOD) samples.
4.	The model uses a functional consistency loss to ensure the counterfactual image produces a desired outcome from the deep classifier.
Finally, the model uses progressive optimization to learn the image prior layer-by-layer.


**Limitations And Societal Impact:**

no significant social concern

**Main Review:**

# After rebuttal

the authors have performed extended experiments to compare against existing work, that has substantially improved their manuscript. So I would like to change my rating from 6 to 7

----------------------------
Major comments
•	In section 3.2, "Choice of Image Priors: Total variation + l2": From equation 4, the initialization of the counterfactual image is not clear.
•	 In section 3.2, "Choice of Image Priors: DIP and INR," the definition of z is not clear. The authors should consider moving text from supplementary material to the main manuscript to clarify how the counterfactual image was synthesized using DIP and INR.
•	 In Eq. 5, the mc-loss term is dependent on the counterfactual image and the deep classifier. In DUQ, the mc-term considers the embedding of the centroid for the target and source class and hence depends on y. Eq. 5 should be updated to show this dependence.
•	In Eq. 8, the definition of s* is not explicit.
•	 In section "Findings: LSO vs. ISO,": only qualitative results in Fig.3 are provided. To compare the quality of the images generated across different models, the authors should consider reporting results on metrics such as FID score.
•	In section "Findings: Manifold consistency is essential,": only qualitative results in Fig.4 are provided.
•	In Table 1: the best scores are not highlighted to show the best performing model.
•	Fig.4 and Table 1 emphasize identifying the model with the most localized and meaningful counterfactual changes. But no results are reported on whether the changes are important/relevant for classification. The authors should consider comparing against post-hoc feature importance-based methods such as Grad-CAM and quantify the intersection between salient regions obtained from different methods. The authors should also consider reporting log odds metric, which is a measure of change in prediction when k% of the most relevant features in the input data are masked.
•	In Fig. 6, the authors assumed the reader know about ABCD signatures for understanding skin-lesion images. An example to describe how perturbations are consistent with this signature will help readers understand the results.
•	In Fig.7, usually, the uncertainty of the classifier increases with shifts in data distribution. It would be interesting to see how certain the classifier is in its prediction for the counterfactual images. Does the proposed method successfully create a counterfactual image with a certain prediction for a query image with high uncertainty?
•	To improve the readability of the figures in the result section, authors should consider reporting f(x) before and after counterfactual changes.
•	Sometimes, a classifier may not use visual clues meaningful to humans to make prediction decisions, i.e., a classifier may rely on spurious features or shortcuts to make prediction decisions. An application of the current method on discovering such shortcuts will improve the quality of the manuscript.
•	Update the references to arxiv versions to point to actual publications.




**Time Spent Reviewing:**

4

---

> ### Author Response · Authors · 2021-08-10
> **Response to Reviewer JzXC**
>
> We thank you for your valuable comments and feedback. We would like to address your concerns and comments as follows:
>
> > **FID Scores to compare ISO vs LSO:**
>
> When viewed from the lens of popular image quality measures from the generative modeling literature, e.g. FID scores, precision and recall, counterfactuals from ISO and LSO style optimization are similar. More interestingly, as shown in our response to Reviewer Txfe (refer Table A), the image quality from DISC is not very inferior to using pre-trained GANs as the image prior. However, in terms of the perturbations introduced, LSO consistently produces more concentrated and discernible changes, while also providing much lower classifier discrepancy (CD) scores (Refer to table below). Hence, we believe we need to collectively analyze multiple metrics to characterize counterfactual optimization methods.
>
> _Table C. Comparing ISO and LSO optimization for CelebA Not Smiling -> Smiling (Results from 5000 synthesized images)_
>
> |Method&nbsp;|FID &nbsp;&nbsp;|MSE &nbsp;&nbsp;|Concentration&nbsp;&nbsp;|Classifier Discrepancy|
> |:-:|:-:|:-:|:-:|:-:|
> |ISO + INR + DEP &nbsp;|**80.6**|0.13 $\pm$ 0.04|0.39 $\pm$ 0.17|0.19 $\pm$ 0.12|
> |LSO + INR + DEP &nbsp;|81.7&nbsp;|**0.22 $\pm$ 0.07**|**0.18 $\pm$ 0.11**|**0.08 $\pm$ 0.03**|
>
> > **Identifying changes relevant for classification:**
>
> This is a very important question. Note that, though DISC is designed for generating counterfactuals for different user-specified hypotheses on the predictions, it can also be repurposed for obtaining saliency maps. We compute the saliency map for a target class as follows: $|\mathcal{C}(x, y=1) - \mathcal{C}(x, y=0)|$, where $x$ is the query image, $\mathcal{C}(x)$ is the counterfactual, $y$ is the target class. We find that saliency maps from DISC are very comparable to standard approaches such as GradCAM (in terms of $\Delta$log-odds). For example, saliency maps for the smiling->not smiling classifier picks the regions near the mouth and cheeks as the influential features.  Due to space constraints in the main paper we will include this experiment to the final version of the supplementary material. To better position our method, we will compare our proposed approach with other SOTA baselines such as DeepSHAP and CXplain. Please find below the comparison to Grad-CAM. Note, the $\Delta$log-odds metric was computed by masking the most influential features (zero-masking) and measuring the change in prediction (higher the better) - $\Delta$log-odds = log-odds($F(\mathrm{x})$) - log-odds($F(\mathrm{x}_{\text{masked}})$), where log-odds(p) = $log(p/1-p)$.
>
> _Table D. Comparing saliency maps of $100$ randomly chosen smiling images from DISC and GradCAM by zero-masking top $15\%$% of influential features and computing $\Delta$log-odds._
>
> |&nbsp;&nbsp;|GradCAM &nbsp;&nbsp;&nbsp;|DISC|
> |:-:|:-:|:-:|
> |$\Delta$log-Odds|10.4 $\pm$ 3.3|11.1 $\pm$ 3.7|
>
> >**Predictions from the classifier for the CFs obtained from corrupted observations:**
>
> Thanks for pointing this out. By reprojecting even OOD test time examples closer to the data manifold (Fig. 2 from the paper), DISC is likely to produce CFs with more confident predictions. Please find below the average softmax probabilities of the corrupted query image P(not smiling|query) and the counterfactual P(smiling|counterfactual).
>
> _Table E. Corrupted CelebA images, not smiling -> smiling. We report the average prediction confidences from 500 images (i.e., P(not smiling|query) / P(smiling|counterfactual)_
>
> |Brightness (sev = 3)&nbsp;&nbsp;|Brightness (sev = 5) &nbsp;&nbsp;|Gaussian Noise (sev = 5)&nbsp;&nbsp;|Motion Blur (sev = 5)&nbsp;&nbsp;|Zoom|
> |:-:|:-:|:-:|:-:|:-:|
> |0.79 / 0.94|0.58 / 0.87|0.84 / 0.96|0.49 / 0.71|0.51 / 0.72|
>
>
> >**Using DISC to detect shortcuts:**
>
> Thanks for pointing this out. DISC can effectively reveal shortcut rules and presence of counfounding factors in the decisions made by black-box deep learning models. We will include this in the discussion section.
>
> >**Notations:**
>
> + While using total variation, we initialize the counterfactual (CF) image as a realization from a normal distribution. In section 3.2, we will better clarify the initialization procedure.
> + In section 3.2.b, Deep Image Prior: The noise $\mathrm{z}$ is of the same dimensions as that of the query image drawn from a uniform distribution $\mathcal{U(-1,1)}$; In section 3.2.c, INR:  $\mathrm{z}$ is the Fourier feature mapping of the 2−D input coordinate grid.
> + We will update Eqn 5 in the final version of the paper to denote the appropriate dependence.
> + We missed to include this in the paper and we will definitely update it in the final version. $s^*$ in Eqn. 8 denotes a target error estimate, we require our synthesized CF to produce. In our experiments we set $s^{*}$ as the median error estimate from DEP on a held-out validation set.
> + We will highlight the scores of the best performing models  in Table 1 of the paper.
> + In Figure 6 we will better highlight the regions in the skin lesion CF images that are directly reflective of the ABCD signatures in the paper.
> + We will also update the classifier scores $F(x)$ before and after the CF changes in all the figures in the paper.
> + We will update the references from their arxiv versions to their actual publications. Sorry for the oversight.

---

### Official Review · Reviewer_Txfe · 2021-07-20

**Rating:** 5
**Confidence:** 4

**Summary:**

The paper proposes a deep inversion approach that uses a pre-trained classifier to generate the counterfactual explanations of a given image. In other words, the method introduces small, discernible perturbations in an image that changes its class while keeping it realistic. The method makes use of certain inductive biases such as strong image priors, manifold consistency objective, and progressive optimization strategy to generate those counterfactual explanations.

**Limitations And Societal Impact:**

None that I know of.

**Main Review:**

Contribution: The method differs from other counterfactual generating methods which use generative models. This method relies on inverting a discriminative model (a pre-trained classifier) to generate counterfactuals which I believe is novel and pretty interesting. Controlling this inversion (manipulating the synthesis part) is difficult and the proposed method provides ways to control it via certain inductive biases.

Main comments/ questions:
- The paper provides ablation for the design choices which is great. However, I am missing the comparisons with the known counterfactual methods such as generative models based methods.

- The results are verified on simpler datasets where the difference in classes requires very small manipulations/ perturbations. Also, the number of classes in these datasets is very limited for e.g. there are only three classes in CelebA. How would the method deal with more complicated datasets such as image net? How do the perturbations look there as the data manifold is more complicated? I would be interested in seeing results on at least a subset of image-net or similar datasets.

Clarity:  The paper is fairly well-written and easy to follow. However, the motivation of the approach is somewhat lacking. For e.g. the authors say that one cannot use previously proposed counterfactual generative models because of privacy constraints. It would be great if they can provide some references. Moreover, I also found the notations to be a bit confusing. I will expand on this below.

- In eq 1, where does $\mathcal{C}$ come from? Is it a subset of $\mathcal{M}$?

- In line 141, what’s z?

- I didn’t understand the role of contrastive objective in preserving the ordering of the samples in DEP (sec. 3.3(a)). And where is this contrastive objective used?

- In the experimental setup, for designing a classifier, why is the gradient penalty and length scale parameter set to 0.5?

I generally liked the idea, but I believe that the paper can be improved further with more results. For now, I am inclined towards borderline rejection.


**Time Spent Reviewing:**

11

---

> ### Author Response · Authors · 2021-08-10
> **Response to Reviewer Txfe**
>
> We thank you for your constructive comments and feedback. We hope our responses address your concerns and you can champion our paper.
>
> > **Comparison with Generative Model based Counterfactual (CF) Generation Methods:**
>
> Though our focus was on test-time CF generation without assuming access to generative models or training data, we generated results for a GAN-based baseline (similar to Sauer & Geiger, 2021). As expected, the image quality is better with pre-trained generative models (as indicated by lower FID, and higher precision/recall scores). However, in terms of evaluation metrics such as the proposed classifier discrepancy (CD), it is not significantly different than our proposed approach (DISC). This clearly demonstrates the ability of DISC to effectively manipulate the most relevant image features and produce representative counterfactuals for the target class. We provide the initial results of this comparison for the case of CelebA dataset (smiling attribute) below and will include more extensive results in a potential final version of the paper.
>
> _Table A. Comparing GAN-based CF generation against DISC. CelebA Not Smiling -> Smiling (Results from 5000 synthesized images)_
>
> |Method&nbsp;|FID &nbsp;|Precision&nbsp;&nbsp;|Recall&nbsp;&nbsp;|Classifier Discrepancy|
> |:-:|:-:|:-:|:-:|:-:|
> |GAN-based &nbsp;|**69.4**|**0.71**|**0.27**|**0.07 $\pm$ 0.04**|
> |DISC&nbsp;|81.7&nbsp;|0.66|0.19|**0.08 $\pm$ 0.03**|
>
> > **Performance with multi-class datasets:**
>
> We have performed experiments in multi-class settings (ISIC 2018 dataset with 7 classes) and Fig. 6 of the paper shows the counterfactuals from a multi-class classifier. Our results show that we are able to transition effectively between different lesion types. Without loss of generality, we expect DISC to work for other multi-class classification problems. Note that, for simplicity, the classifier discrepancy metric reported in the paper was based on a binary classifier with Melanoma and Melanocytic nevus lesion types (_Table 1 in the paper_). To demonstrate that this observation holds even with multi-class classifiers, we repeated the experiment by replacing one target class at a time using counterfactual images and computed classifier discrepancy for the multi-class classifier. Please find the results below, which we will include in the final version of the paper.
>
> _Table B. ISIC 2018 Multi-class Classifier. Original images in each target class are replaced using counterfactuals (1200 images) from the remaining classes (~200 from each class) and the classifiers are retrained. Results are from three classifiers trained with different random seeds._
>
> |&nbsp;&nbsp;|Vascular lesion|Melanoma|Basal cell carcinoma|
> |:-:|:-:|:-:|:-:|
> |Classifier Discrepancy|0.08 $\pm$ 0.02|0.06 $\pm$ 0.01|0.03 $\pm$ 0.02|
>
> > **Motivation for DISC:**
>
> CF generation is a challenging problem requiring the synthesized images to be realistic and close to the original image manifold. Existing approaches adopt pre-trained generative models or assume access to the training data distribution for CF synthesis which are known to easily satisfy the above criteria. However, in practical scenarios such as in healthcare, one may often have access only to a classifier pre-trained on a target dataset which is not publicly accessible. In such settings, one cannot access the training data directly or indirectly in the form of generative models for CF synthesis. Although approaches such as Deep Inversion have been proposed to synthesize images using only a classifier for tasks such as data-free knowledge distillation, they are not sufficient for generating CFs. In this context, we propose to design test-time data-free CF generators for an arbitrary classifier. We will include additional relevant references for motivating this in the final version of the paper.
>
> > **Role of the Contrastive Objective in DEP:**
>
> To obtain a reliable direct error predictor (DEP) that is consistent with the uncertainties of the classifier, we first jointly train the classifier and error predictor by optimizing a weighted combination of the cross-entropy loss and the contrastive loss. The contrastive loss by design makes the DEP discard the overall scale changes to the classifier losses, and enforces only the relative ordering of the samples based upon the error from the classifier. Upon training, the DEP can then be used to identify the regimes of low and high model confidence, for example, to distinguish OOD and in-distribution samples. Although the DEP can directly regress the true losses from the classifier by using a simple MSE objective, we find that it leads to poor convergence (trivially producing average losses). Note, this contrastive objective is used only during the joint training and not while generating CFs. We will clarify this in the paper and include comparison between convergence of MSE loss/contrastive loss in the supplementary.
>
> > **Length Scale and Gradient Penalty:**
>
> These hyper-parameters are associated only with the DUQ baseline.  Based on a hyper-parameter search, we found that a length scale and gradient penalty parameter of 0.5 to produce best performance (similar values were reported for CIFAR-10 and SVHN benchmarks in the original DUQ paper).
>
> > **Notations:**
>
> + In Eqn. 1, $\mathcal{M}(.)$ and not $\mathcal{M}(x)$ denotes the image manifold while $\mathcal{C}(x)$ denotes the counterfactual of the query image x.
> + In line 141, for deep image priors, the noise $\mathrm{z}$ is of the same dimensions as that of the query image drawn from a uniform distribution $\mathcal{U(-1,1)}$.

---

### Decision · Program_Chairs · 2021-09-27

**Decision:**

Accept (Poster)

**Comment:**

The reviews highlight that this a technically sound, well-written contribution to the problem of counterfactual explanations in the novel and relevant setting where only the classifier is accessible. The manuscript is therefore accepted. The impact of the paper could be further strengthened if it was put in context of other ongoing work of counterfactual explanations, eg [1].

[1] Karimi et al. A survey of algorithmic recourse: contrastive explanations and consequential recommendations